# The Third Skin: A Biomimetic Hydronic Conditioning System, a New Direction in Ecologically Sustainable Design

**DOI:** 10.3390/biomimetics10120843

**Published:** 2025-12-16

**Authors:** Mark B. Luther, Richard Hyde, Arosha Gamage, Hung Q. Do

**Affiliations:** 1Environmental Energy Services, Geelong 3224, Australia; mark.b.luther@gmail.com; 2School of Architecture, Design and Planning, University of Sydney, Sydney 2008, Australia; richard.hyde85@gmail.com (R.H.); arosha.gamage@sydney.edu.au (A.G.); 3School of Architecture and Built Environment, Deakin University, Geelong 3220, Australia

**Keywords:** hydronic, buildings, conditioning systems, biomimicry, bioinspired, sustainable ecological design

## Abstract

The increasing demand for sustainable climate control has spurred research into our hydronic conditioning system with a patented radiant ceiling panel (AU 2024227462) inspired by biomimetic methodologies. This study develops a framework that utilizes natural systems for heating and cooling, enhancing system performance and environmental sustainability. Biometric analysis was the primary method for testing these systems, focusing on heat transfer mechanisms modeled after human biology. Findings indicate that the proposed hydronic system excels in cooling mode, achieving an average capacity of 95 W/m^2^ while maintaining thermal comfort levels (PMV) with solar heat gains under 1.5 kW in an 18 m^2^ space. However, in heating mode, the system shows a capacity of 85 W/m^2^ but struggles with vertical air-temperature stratification, especially in the radiant ceiling component. This highlights the potential of biomimetic designs to enhance energy efficiency and comfort in sustainable development. The hydronic panel system parallels the human body in energy transfer; both can emit 75–90 W/m^2^ through radiation. Convection over the panel can increase energy transfer by 50–80%, akin to the human body’s heat loss through convection. Notably, natural perspiration facilitates latent energy transfer of 20–25%. When the conditioned panel operates below the dew point, it generates water vapor, boosting cooling capacity by 5–15% and enhancing latent energy transfer. Overall, the heat transfer processes of the hydronic panel mimic certain aspects of human physiology, distinguishing it from conventional HVAC systems.

## 1. Introduction

Hydronic radiant systems are conditioning systems tasked with providing thermal comfort (heating or cooling), similar to the traditional air conditioning system. However, while the air conditioning system uses convection as the primary heat transfer mechanism, blowing conditioned air into the spaces that require conditioning, the radiant system mainly utilizes radiant heat transfer [1]. While the air conditioners control the air temperature, air velocity, and air humidity, the radiant system controls the mean radiant temperature (MRT) and air temperature [2].

There are three main types of radiant conditioning systems, namely Thermal Active Building Systems (TABS), Embedded Surface heating and cooling Systems (ESS), and Radiant heating and cooling Panels (RP). Here, the TABS with water tubes buried in thick and heavy concrete slabs are the early radiant system, used for heating and infamous for being slow and difficult to control [3]. In contrast to traditional, slowly reactive TABS, the ESS and RP are newly developed lightweight systems that are fast and dynamic, instantly responding to a space’s conditioning requirements, and easier to control [2].

Thus, the lightweight and responsive radiant panels, usually used for radiant ceilings, are gaining popularity. There are established systems on the market from major companies such as Armstrong, Messana, Uponor, and Rehau. Also, recently, a huge number of studies have been conducted to improve the thermal and energy performance of the radiant conditioning ceiling system.

For example, researchers have explored alternatives in hydronic systems to augment their output capacity as well as resolve anticipated problems of condensation when cooling. Over the past decade, state-of-the-art material technology for radiant panels has taken place [4]. Phase change materials are used for enhancing cooling output (Gallardo and Beradari) as well as by [5] to absorb and store energy. Yet, these systems encountered difficulties, lacking response and predictive control. On the other hand, Zhang, et al. [6] created panel designs applying radiant cooling fins to increase convective heat transfer, while [7] proposed a ‘grooved radiant panel’. Both of these designs were considered unconventional to the traditional flat ceiling plane, lacking the desired architectural aesthetics.

Approaches to the reduction, or elimination of condensation go back six decades ago when Morse [8] invented the membrane-covered radiant panel design. Morse’s studies have been recently revisited and revised by Gu, et al. [9]. However, while the membrane isolates the condensation problem from the room, it also reduces the cooling capacity of the system. Other researchers, such as Zhong, et al. [10], Liu, et al. [11], and Jiang, et al. [12], have looked into creating superhydrophobic material through the application of high-end nanotechnology. In all of these cases, the superhydrophobic material is applied to the finishing surfaces to prevent condensation. The material allows for the surface-controlled temperature to drop below the dew point, increasing the output capacity of cooling. Yet, these materials provide a challenge in manufacturing and cost reduction.

Recently, the research of Bouacida, et al. [13] has considered more practical solutions for radiant cooling panels based on their low cost and effectiveness. They are advocates for ‘primitive’ systems that can provide acceptable results. Similarly, our research has been cost-driven in search of practical solutions and ‘off-the-shelf’ materials that can perform optimally [14].

Overall, the question of how to further improve the performance and the capability of providing thermal comfort of the radiant conditioning system remains. In response to this past research, none of the papers found make any reference to their designs as inspired by nature. None explores the conditioning mechanisms of heat transfer (radiation, convection, evaporation) as they would occur in natural organisms or in nature itself. The concepts of manmade systems reside within themselves and are faced with internal problem-solving solutions. It is therefore that this paper makes a reference to hydronic circulatory conditioning systems as well as their potential heat transfer mechanisms with those of nature through the human vascular system.

Biomimicry is a study of nature to make a comparison of the man-made object with that of nature, and study the principles of nature so that the man-made systems can be improved. The scope of Biomimetics leverages biologically derived mechanisms to produce engineering solutions [15]. It also includes pedagogical methods that can advance the ability of designers, engineers, and scientists to identify and extract biological mechanisms for use. The work identifies a potential biomimetic design, evaluates its utility, and increases our capacity for biomimetic design processes [16]. In this study, the objective is to use biometric methods to provide insight into potential improvements of the radiant conditioning systems and a testing methodology to meet new challenging situations. 

While this may be the case, engineering solutions for the past 100 years have primarily focused on forced convective solutions through HVAC and split systems. Disengaged from any radiant conditioning. Therefore, it may be essential to revisit natural systems in pursuit of technological innovation in building conditioning. Realizing how optimal spatial conditioning might occur or the physical concepts to achieve it is a good start. Ingrained in this journey, biomimicry itself might lead to a better understanding and/or solution.

Finally, exploring technologies that can offer, as much as possible, the physical mechanisms that our human body engages with might be in the right direction. A reversed standpoint from traditional thinking is to ask how the human body loses or gains energy to maintain a balance of comfort. New and emerging challenges arise for the use of hydronic radiant cooling in Aboriginal communities in Australia. Traditional community buildings use open shelters combining radiant, evaporative, and air movement to service these buildings.

While it is of interest to consider biomimetic analysis of a conditioning system for our environments, there may be a greater purpose and significance in doing so. Innovations that are inspired or analogous to nature and natural systems often take less effort, operate more efficiently, and accomplish greater results.

The purpose of the study is to test this hypothesis, an implementation of the biomimetic method in designing a radiant conditioning system for improved thermal performance. Via the biomimetic method, principal design solutions for an improved radiant system are proposed. To support the design solution, the empirical results from our previous experimental studies are referenced. The contribution of this paper is mainly theoretical, focusing on the biomimetic method, which can be implemented in designing radiant conditioning systems. The presented radiant ceiling system was tested in previous studies and proved to be a potential solution. This paper, however, will summarize and highlight the implementation of the biomimicry method in designing a radiant conditioning system. The component, function, and operation of the human circulatory system are compared with those of a hydronic radiant system. Logical arguments will be provided to suggest the most pertinent designs based on biometric analysis.

Also, in recent research into how to improve and renovate buildings in Australia effectively, there was a consensus that the ceiling or roof element held the most significant potential [17,18], and a radiant conditioning ceiling system could offer a solution. Numerous recent papers highlight innovative approaches to thermal comfort that emphasize both energy efficiency and user satisfaction. For instance, Smith, et al. [19] explored integrating innovative control systems into residential heating systems to optimize energy consumption while maintaining user comfort.

In addition to these contemporary findings, an underrepresented yet fascinating technology in the literature is wall heating panels with heat pipes. This radiant heating system combines heat transfer mechanisms, as discussed in the introduction, which discusses various forms of heat transfer. The principle involves extracting heat from water by evaporating the working fluid in heat pipes, with the panel’s surface heated by the vapor’s condensation. Remarkably, this mechanism has parallels in nature. For example, the process of transpiration in plants follows a similar principle: moisture evaporates from leaves, cooling their surfaces and affecting the surrounding microclimate. Additionally, thermoregulation in particular animal species demonstrates that evaporation plays a crucial role in heat management. For instance, many reptiles use behavioral adaptations to control their body temperature by utilizing sun-exposed areas and shade. Identifying these natural mechanisms enriches the discussion but also draws intriguing parallels between engineered systems and nature. This approach emphasizes the potential of bio-inspired designs to yield innovative solutions for sustainable heating technologies.

## 2. Methodology: Biometric Analysis

The proposed radiant system is designed with biomimetic inspirations. This section will explain the fundamentals of designing with biometric analysis in the proposed radiant conditioning system.

### 2.1. Bio-Inspired Ecological Sustainable Design Innovation

Sustainable design focuses on optimizing building performance while reducing negative impacts on both occupants and the environment. Designers, therefore, integrate sustainable design and energy efficiency principles into our construction and modernization projects. This approach balances costs with environmental, social, and human benefits, helping to achieve an agency’s mission objectives and functional needs. Sustainable design principles aim to [20]:Optimize site potential.Minimize non-renewable energy consumption and waste.Use environmentally preferable products.Protect and conserve water.Improve indoor air quality.Enhance operational and maintenance practices.Create healthy and productive environments.

Sustainable design incorporates ecological considerations, adding a focus on conserving natural systems. Further extending this concept with biomimetics emphasizes the importance of nature in developing innovative adaptations in buildings; it is an immersive process (Figure 1). This model implies that the process of an organism is how its primary and secondary functions integrate, with the ability to adapt to the local climate and context that initiates the form and shape that fits the ecosystem [21]. The concept model of Biomimicry is described in the study of Gamage and Hyde [22].

### 2.2. Role of Biomimicry in System Design

In biomimicry, a design mimics the strategies of an organism, a behavioral pattern, or a system in nature. Within the biomimicry approach, there are two means of understanding, namely Direct and Indirect sources from natural systems [16]. Table 1 shows the two ways of understanding the biomimicry approach [22].

Direct sources from natural systems: Conceptualizing Biomimetics as problem- and solution-driven approaches can enhance design. While the problem-driven approach involves the designer seeking to develop a solution to a problem through biological principles, the solution-driven approach consists of emulating biological solutions and transferring them to human design systems.

Indirect sources from natural systems: The indirect approach to biomimicry emphasizes abstraction by characterizing the functionalities of natural systems. Numerous authors and practitioners have compiled essential environmental attributes from natural systems used to develop design principles, lessons, and concepts inspired by characteristics found in the natural world.

Regardless of the chosen approach, biomimicry advocates for three escalating levels of sustainability derived from the shapes of living organisms, the manufacturing processes followed by these organisms, and the interactions within species, ultimately reflecting the global functioning ideal fit to its respective natural ecosystems [22]

### 2.3. An Explanation of the Biometric Analysis

A central feature of the biomimetic analysis is analog translation. Several systems can be used (See Table 2)

The basis of biometric analysis is the use of analogical translation systems. There are several approaches (see Table 2). The basic principles of the analogical approach are used and adapted to the project.

**Scales of application:** How is the biomimetic-inspired hydronic conditioning system integrated with architectural facade systems in buildings?**Ecosystem:** How does the integrated system fit with its environmental context?**Process:** How does the integrated system perform?**Form:** What are the physical characteristics of the systems?**Biomimicry design approaches:** Direct sources from natural systems (Bio-inspiration) and Indirect sources from natural systems (Third skin).

### 2.4. Steps for Conducting Biometric Biomimetic Analysis for Hydronic Systems

There are three steps in conducting this biometric biomimetic analysis for designing hydronic systems:**The scale of application:** process, form, ecosystem levels of the functioning of the human body, and its analogical transfer to the hydronic system.**Design process:** categorization, functional integration, environmental adaptation, and expression of the form of both systems.**Biomimicry approach:** direct (mimicking the circulatory system of the human body) and indirect approach (design strategies, lessons, and principles taken from the third skin strategy).

### 2.5. Implementing Biomimicry in Radiant Conditioning System Design

As shown in Table 1, there are two main approaches to implementing biomimicry, namely direct and indirect inspiration. Both approaches are utilized in analyzing the subject radiant conditioning system. The proposed system is directly inspired by the human circulatory system and heat transfer mechanisms, while indirectly inspired by human skin. Comparisons are made between the biomimetic inspirations and the components, design, and function of the proposed radiant system.

## 3. Direct Inspiration from Human Thermal Comfort and Heat Transfer

### 3.1. Heat Transfer and Thermal Comfort

The hydronic radiant system is directly inspired by the way the human body naturally transfers heat to the surrounding environment. It is helpful to consider natural mechanisms of heat transfer and how human bodies exchange energy within our environments to maintain comfort. Figure 2 illustrates the magnitude of the impact different heat transfer mechanisms have on contributing to body conditioning. Radiative exchange has the most significant influence. 

As shown in Figure 2, designers often raise questions about whether the convective principle, now widely applied, is the correct form of conditioning and how the human body conditions itself in establishing an equilibrium temperature. Radiation alone accounts for close to 50% of our heat transfer exchange within our environment. Together with convective exchange, these two mechanisms make up approximately 75% of the heat transfer within an environment. Therefore, it is common sense to engage with conditioning systems that could regulate and control these two mechanisms of heat transfer within a space.

From a biomimetic viewpoint, humans function to lose or gain energy to maintain body core temperature. In the past, people used forced ventilation to cool or heat their bodies. This air blowing over the body or present in the room environment is taking away or providing energy to the occupant. It is air alone that either cools or provides heat. Meanwhile, there is a lack of consideration of other surfaces in the room that are also absorbing this energy. The human body is competing with these unconditioned surfaces in trying to maintain equilibrium with its environment [24].

A commonly held viewpoint points out that “Buildings don’t use energy, people do,” and a significant proportion of such energy provides thermal comfort [25], leading to the conclusion that “buildings don’t need conditioning, people do”.

At the beginning of the design of spatial conditioning, designers needed to recognize that there are several alternatives. First, the environmental conditions should be evaluated. Second, with split systems as well as HVAC ducted conditioning systems, an air temperature thermostat according to a set-point temperature regulates the system. Third, whether or not thermal comfort has been reached or exceeded, what the thermostatic temperature setting conditions are, and whether the space to the set-point has been accomplished.

Interestingly, while a radiant system can overcool or overheat a space, it involves an aspect of self-regulation via convection as well as radiation. Natural convection, as well as radiation, are mechanisms of heat transfer within a space, are driven by the magnitude of the temperature difference between two objects (surface to air or surface to surface). The implication is that when there is a greater temperature difference between a conditioned surface and its environment, there will be more energy transfer. As the surfaces, as well as air temperature, approach the regulated surface temperature (never reaching it, but closer in temperature difference), the energy exchange from the conditioned surface reduces automatically in both radiative and convective transfer. In contrast to this, conventional air-conditioning does not ‘back off’ automatically and continues to aimlessly condition air, often beyond requirement, hence wasting energy. 

Owners should view a radiant system differently; the heated or cooled surfaces allow the body to regulate its temperature [24]. A bioinspired radiant system is a different paradigm from an air-conditioned environment, which attempts to provide conditioning to the body. We can think of planar radiant surfaces as systems that facilitate our body to give off or receive heat. Of course, these radiant systems will eventually provide convective conditioning as well. It is often, however, recognized in our empirical experiments (in cooling) that MRT can be 3–4 °C lower than air temperatures. Occupants can sense comfort while air temperatures are still high; MRT is more representative of human thermal sensation than air (Dry Bulb) temperature [26].

More experimentation and understanding are required here in the future. It is highly possible that more exploration into biomimetic examples could assist in a better understanding of how nature itself deals with temperature differences and extremes to maintain comfort equilibrium. 

### 3.2. Adaptive Thermal Comfort Model

Currently, the PMV (Predicted Mean Vote) and the adaptive thermal comfort model are the two most widely used thermal comfort models internationally [27]. Here, the PMV model is based on four objective variables and only two personal variables. The PMV model is mainly used for conditioned closed interior spaces and is usually considered conservative [28]. Meanwhile, the adaptive comfort model can be considered more natural and is based on the adaptability of the human body to its thermal environment [27]. Occupants have more control over the thermal environment via actions such as opening windows, wearing suitable clothing, and even adjusting their metabolic rate (activity) to gain thermal comfort. Hence, adaptive thermal comfort follows closer to the biological response of the human body. Thus, occupants can accept a temperature range [29]. The key variable in the adaptive comfort model is the indoor operative temperature, representing the temperature that people actually feel. The optimized indoor operative temperature can be calculated based on the prevailing mean outdoor temperature. To develop an automatic control mechanism for the radiant ceiling system, the adaptive thermal comfort model has been adopted.

## 4. Direct Inspiration from the Human Circulatory System

### 4.1. Human Circulatory System Components and Key Processes

There are several biomimetic analogies between the human circulatory system and the hydronic operations distribution system. While a pump may be analogous to the heart and a pressure vessel or a fluid temperature mixing valve with other components in the human vascular system, these are probably not the most critical or significant aspects of a biomimetic innovation analogy.

The Human Circulatory System includes:**Heart:** The heart is a muscular organ that pumps blood throughout the body, divided into four chambers: the left and right atria, and the left and right ventricles.**Arteries:** Arteries are blood vessels that carry oxygenated blood away from the heart to the rest of the body. They are thick-walled and muscular to withstand high blood pressure.**Veins:** Veins are blood vessels that carry deoxygenated blood back to the heart. They have thinner walls than arteries and are less muscular.**Capillaries:** Capillaries are tiny blood vessels that allow for the exchange of oxygen and nutrients with tissues and organs.**Blood:** Blood is a liquid tissue that carries oxygen, nutrients, hormones, and waste products throughout the body.

Key Processes:**Systole:** The heart contracts to pump blood out of its chambers.**Diastole:** The heart relaxes between beats, allowing blood to flow back into its chambers.**Pumping Action:** The heart pumps about 2000 gallons of blood every day.**Blood Pressure Regulation:** Blood pressure is regulated by adjusting blood vessel diameter, which affects blood flow. This aspect is similar in a hydronic system, where tubing diameter changes flow rate and pressure.

### 4.2. Blood Vessels and the Piping in Radiant Systems

The concept of fluid distribution within a hydronic system is analogous to the human body. Arteries and large-diameter veins are the main supply and return distribution piping within the human body, mimicking the supply and return piping in the layout of a hydronic system. The arteries carry blood pumped from the heart to the tissues. Chemicals are exchanged via capillaries, and the veins carry them back to the heart [30]. While the blood circulatory system delivers nutrients and oxygen from the heart (the pump), it also provides a return of deoxygenated fluid (blood) deprived of nutrients back to the start of the system. During this process of supply, the capillaries play a vital role in the delivery and distribution of oxygen and nutrients to the most extreme parts of the body [30]. In the heat exchange between the human body and the surrounding environment, the capillaries of the human skin play the most critical role [31].

It is only natural to compare hydronic systems with our human vascular system. Interestingly, the blood flow velocity in the human body differs depending on the type of vessel [30]. Human vessels are flexible and active (alive), and their compliance and diameter vary according to hormonal and nervous system inputs, so the capacity and pressure of the venous reservoir can constantly change. Likewise, the arteries and small arteries (arterioles) have muscle in their walls, but much more than veins, as they must withstand higher pressure. They can constrict and change their resistance to flow all the time.

Human capillaries have a highly selective permeability, so substances can cross without the blood or necessary blood constituents leaking out. The analogy would be with insulation in the central pipes versus heat-conducting material in the capillary network.

Human blood pressure is controlled beat by beat using neural mechanisms acting on muscle in arteriole walls and over extended periods by mechanisms that control blood volume. Because of the elasticity and contraction of the vessel walls, our circulatory pressure swings are much more damped than in a solid pipe system. Various mechanisms prepare the circulation for action, fight, and flight by bumping up the pressure and flow in advance.

A variety of vessel sizes that allow for changes in pressure regulate the flow rate. Similarly, both systems consist of a network of tubes with varying diameters that help control the flow rate, resulting in pressure differences throughout the system.

In the human system, arteries supply and ‘conditioned fluid’ (blood), which are thick-walled ‘tubes’ that can withstand a higher pressure than the veins. Similarly, capillary hydronic systems have a ‘pressure’ system maintained throughout their closed loop, generally at about 2.0 bar or less.

### 4.3. Heart vs. Hydronic Control and Distribution System

Both the human body and a hydronic system are examples of ‘closed loop’ circulatory systems; both operate on a supply-and-return concept for the fluid. In both systems, the flow of liquid is directional; the human body delivers the fluid by the heart, while in a hydronic system, it is delivered by a pump.

It is essential to maintain uniform and constant pressure throughout the hydronic system over time with a pressurized vessel that contains a bladder. This vessel must operate effectively across various temperature changes when transitioning from hot to cold conditions. The rubber bladder, designed similarly to a balloon, performs this function within the pressurized tank.

In humans, there is a similar analogy: the venous system acts as a low-pressure reservoir that supports the high-pressure side of the system (the arteries), much like the pressure bladder does in the hydronic system. The human body can adjust pressure in response to different conditions, ensuring efficient circulation and function. Our circulatory pressure swings are much more damped than in a solid pipe system, because of the elasticity of the vessel walls. Various mechanisms prepare the circulation for action, fight, and flight by bumping up pressure and flow in advance.

### 4.4. Capillary Tube Mat and Capillary Blood Vessels

In the radiant surfaces of modern hydronic radiant systems, there are two types of piping. The traditional piping system uses serpentine, 10 to 20 mm in diameter, metal or plastic tubes to distribute heat to the radiant surfaces (walls, floors, or ceilings). This piping system is simple and can ensure high flow rates, but it can also result in uneven heat distribution. Capillary piping systems are one of the newest concepts in hydronic space conditioning. A capillary tube mat consists of two manifold tubes (about 20 mm in diameter), one connected to the water supply piping and the other connected to the return. The two manifold tubes relate to each other by a series of parallel capillary tubes (about 3–4 mm in diameter). The capillary tubes exchange heat with the radiant surfaces and then the conditioned room. In comparison with the traditional serpentine tube, the capillary piping system offers a more even and efficient heat distribution [14,32]. Also, capillary tube systems provide more flexibility in system design.

Capillary networks provide an effective means of functioning in both heat transfer systems (i.e., surface area) for exchanging radiative and conductive energy (Figure 3). In the hydronic case, it is heat, but in the human body, it is heat, nutrients, oxygen, and CO2, as well as other waste products, which are recycled. The heat exchange occurs in the skin, making it the closest analogy to hydronic heating systems.

Of course, there are some distinctive differences in operation between the human circulatory system and a hydronic system. While oxygenation of the blood is a vitally important part of the human circulatory system, oxygenation within capillary tubing is problematic. It is a constant battle within a hydronic system to release oxygen from its circulatory system. One method for optimizing a hydronic system is through the pressurization of the tubing, which increases the air pressure within a vessel and also raises the pressure in the closed-loop circuit. As mentioned, pressurization plays a crucial role in the ‘bleeding’ process during the initial stages of commissioning the system. The goal of this bleeding process is to remove as much oxygen as possible from the tubing, as air bubbles can obstruct the flow and efficiency of the system

### 4.5. Hydronic Radiant System Function—Lessons Learn from the Human Circulatory System

Human Circulatory System Functionality:Blood Flow: Blood flows through the circulatory system in one direction, from the heart to the rest of the body and back to the heart.Deoxygenated Blood Return: Deoxygenated blood returns to the heart through the superior and inferior vena cava (large veins).Oxygenation: The heart pumps deoxygenated blood into the lungs, where it picks up oxygen from inhaled air.Oxygen-rich Blood: The oxygen-rich blood returns to the heart through the pulmonary veins and is pumped into the aorta (the largest artery).Distribution: Various parts of the body, through smaller arteries, capillaries, and veins, distribute oxygen-rich blood.Waste Removal: Deoxygenated blood carries waste products back to the heart, where they are removed from the body through exhalation (breathing out).

Importance of the Circulatory System [33]:Oxygen Delivery: The circulatory system delivers oxygen to cells, enabling energy production.Waste Removal: It removes waste products from cells, preventing their accumulation and toxicity.Temperature Regulation: It helps regulate body temperature by circulating warm or cool blood as needed.Hormone Transport: It transports hormones produced by endocrine glands to target cells.

In a hydronic radiant system, conditioned water (chilled or heated) is pumped and distributed to the tubes integrated into the interior surfaces, such as floors, ceilings, and sometimes walls [32]. The temperature of such interior surfaces is changed accordingly, triggering radiant heat transfer followed by convective heat transfer between the conditioned surface and the room. The heat transfer is between the conditioned room and the water inside the tubes through the materials of the interior surfaces, which can be concrete, metal, gypsum, etc., depending on the design of the radiant systems. Here, the operation of the hydronic heating and cooling systems is not too dissimilar to the concepts of conditioning our human bodies through our circulatory system, in which the heat transfer is between the blood and the surrounding environment through flesh and skin. The biomimicry analogy conditions a body and its surface into one that conditions a surface and its space, achieving heat transfer through a fluid circulating through or near its surfaces. The heat transfer through surfaces is an important analogy here.

In a hydronic radiant system, the suggested operating strategy is to pump water at a fast and steady flow rate to ensure adequate heat distribution to a large area of radiant surfaces [34]. Similarly, in the resting stage, a typical human heart provides blood circulation in the human body with a high flow rate of 5 to 6 L per minute [35]. The purpose is to maintain a uniform and balanced temperature within their ambient environment. The heartbeat will increase for faster blood flow at higher metabolic rates to increase blood circulation and enhance heat transfer, activating mechanisms such as sweating, which improves heat transfer between the human body and the surrounding environment [31].

A hydronic system delivers the conditioning fluid to condition a surface at a particular temperature in a room. The human circulatory system also provides for temperature regulation of the body. In addition to this, both systems break down into smaller distribution vessels, like the manifolds that serve a multitude of capillary tubing in the hydronic system. Similarly, capillary vessels ensure the distribution of blood circulation to the remote parts of the body to maintain a constant body temperature.

Of course, the blood circulatory system does much more, being the precedent in nature, than the mimicked hydronic capillary system. The blood delivers nutrients and oxygen to all cells in the body and, in doing so, feeds the tissues and organs of our bodies. It also removes waste products and is, therefore, a highly complex network of organs and vessels.

In addition to central control systems and mechanisms, the human body has local mechanisms. If tissues become low in oxygen or accumulate CO_2_ or acid waste, there is a direct local effect on the incoming blood vessels that dilates them and allows more blood flow into the area. Injured or inflamed areas receive more blood by similar local mechanisms (plus local hormones and transmitter substances mimicking in hydronic systems).

Various special regional circulations in the body may be controlled a little differently depending on their function, e.g., kidney blood flow, coronary vessels in the heart, brain, and liver. The brain circulation is protected in an emergency, even when other regions are closing because of severe bleeding/trauma. Could a hydronic radiant system need different mechanisms or settings for different rooms?

## 5. Indirect Inspiration—The Third Skin

In the heat exchange between the human body and the surrounding environment, the capillaries of the human skin play the most critical role [31]. Hence, the analogy of building surfaces (or skins) can provide for our human conditioning. It may be, however, of greater interest in how the planar hydronic surface is or can become a living or temperature-responsive skin (surface), which requires further explanation. Ultimately, it is the recognition that it is our skin or a surface that provides the interface of conditioning with its surrounding environment. 

An analogy of our human regulatory system at our skin might be helpful. Skin circulation opens when humans are hot and shuts down when we are cold, controlled by arteriolar resistance via the nervous system. Humans have temperature sensors and a feedback system in the brain stem (hypothalamus gland) to maintain body temperature, but it is complicated. Yet, the skin offers evaporative (sweating) conditioning as a form of cooling, as well as a response to radiant and convective conditioning. Similarly, our patented hydronic panel (AU 2024227462) offers a granular-coated substrate (skin) which provides for sharp and uneven edges at its surface that break up water droplets from forming, mimicking the hair on our skin, allowing for a film of water to form. This water vapor will be evaporated through air movement, providing additional cooling to the body, or in this case, the space.

For cooling, ventilation is commonly used in forms of various forms of fans, or simply opening the window. The principle here is that the increased air velocity can enhance the heat transfer between the human skin and the external environment. As the “third skin”, the radiant system can also benefit from the same principle. Interestingly, the heat transfer between the radiant system and the conditioned environment consisted of radiant and convective heat transfer. While the radiant heat transfer coefficient is quite stable at about 5.5 W/m^2^K [36], convective heat transfer can vary. Hence, enhancing the capacity of the radiant system with mechanical convective aid has been proposed by Causone, et al. [37], Pantelic, et al. [38], or Bauman, et al. [39].

## 6. The Infrastructure for the Bioinspired Hydronic System

Control systems for heating, ventilation, and air-conditioning (HVAC) facilitate both rapid and gradual adjustments to achieve the desired set-point conditions. The effectiveness of the response in reaching the targeted thermal environment depends on several factors, including the mechanical design, the quality of the components, the sensors used, and the control algorithms implemented. In hydronic systems, various mechanical components work together within a unit known as the hydronic operations distribution module (HODM). Figure 4 illustrates the HODM system designed to service two distinct conditioning areas (zones), such as floors or ceilings.

The HODM, as shown in Figure 4, has two distinct and separate conditioning modules. Each module consists of a pump, a supply mixing valve, a supply flow rate meter and temperature sensor, a pipe system pressure vessel, a pressure gauge, and a hot or cold tank supply switching actuator. Other measurement sensors essential to the HODM include ambient and internal dry-bulb (air) temperature, room humidity, and, probably most importantly, the ceiling surface temperature. In the future, it is also desirable to include an air velocity sensor in the space.

Figure 5 shows the diagram of a radiant conditioning system with the HODM as the center of operation. Here, the two tank stores exchange heat with the water in the radiant system, similar to the two lungs. The HODM pump and distributed conditioned water like the heart. The conditioned water is supplied to the radiant surfaces (ceiling and floor), which are similar to our skin.

The inputs and outputs to and from the controller of this system allow for a desired agenda to be programmed. There is no doubt that many possibilities exist for controlling the hydronic system to achieve the desired thermal comfort result. Since our patented hydronic panel (AU 2024227462) offers an entire planar and surface conditioning, the Mean Radiant Temperature has a distinctive influence on the thermal comfort of the space.

Due to the influence of mean radiant temperature on the space, through hydronic conditioning, it was desired to pursue the concept of controlling for thermal comfort. Hence, making the conventional idea of a thermostat located on a wall determining the entire control of a space obsolete. The objective now is to develop a ‘comfort stat’ that mainly relies upon the space’s operative temperature [40], as well as other measures that determine thermal comfort. The operative temperature includes the dry bulb and mean radiant temperature. Generally, the two measurements have equal weight, resulting in an Operative Temperature. 

According to Fanger’s static thermal comfort model, several factors influence thermal comfort, including mean radiant temperature, dry-bulb temperature, air velocity, humidity, clothing, and metabolic rate (our level of activity) [41]. A standard view has emerged in the scientific community that air temperature alone does not determine thermal comfort in a space. The human body responds to three distinct mechanisms of heat transfer in the environment, as previously illustrated in Figure 2.

Our control, at this point, may not be as sophisticated as we would like, but it nevertheless follows the principles towards the development of a ‘comfort stat’ based on the previously explained adaptive comfort model. Experimentation has demonstrated to us that different construction compositions of a hydronic ceiling panel will result in different surface temperatures. For example, when 10 °C water temperature is supplied, it will yield a +2.5 °C higher (than supply) surface temperature for a rendered panel, about a +4.5 °C higher (than supply) for a plasterboard panel, and about a +3.5 °C higher (than supply) for a metal perforated panel. Naturally, there would be a temperature sensor on the ceiling to determine the surface temperature result.

Research questions are emerging as to how biomimetics can further assist with how much and where the pressure drops across a capillary hydronic system. It starts at 2 Bar but drops as it returns to the pump. Human arterial blood pressure is pulsatile, but the mean is about 80 mmHg/0.1 Bar as it leaves the heart; the most significant pressure drop is at arterioles, so pre-capillary. Venous blood has dropped to only 5 mmHg/0.006 Bar at the inlet to the heart. The pump mechanics are very different—pulsatile versus continuous; the stroke of the pump changes with the rate and degree of filling of the heart. 

### 6.1. Capillary Flow Systems

The tubing or structures found on plant leaves, which are part of the plant’s vascular system and are responsible for transporting water, nutrients, and food, are called veins. These arrangements of veins in a leaf are called venation, and there are different patterns [42], including:**Parallel venation:** Veins run parallel to each other, often found in monocots like grasses.**Reticulate (or pinnate) venation:** Veins form a network or branching pattern, typical in dicots like broad-leaved plants.**Palmate venation:** Several main veins radiate from a central point, also common in dicots.

The veins of a plant can have a two-fold function, namely, structural support as well as the transportation of fluid and nutrients. Similarly, the molded structure of the aluminum pan, together with the capillary tubing in our patented hydronic panel (AU 2024227462), creates part of the structure as well as the transport of the conditioned fluid (Figure 6). In particular, the aluminum pan serves as a conductor for the conditioning plastic tubing, providing a uniform surface temperature. This results in a very effective transfer of energy between the finishing surface (a granular-aggregate) and the spatial environment.

### 6.2. Serpentine Tubing vs. Capillary Tube Mats

The tubing connection and water flow configuration of our panel are straightforward when compared to conventional serpentine pattern designs (see Figure 7A,B below). As a result, capillary tube mats provide higher heat transfer, uniform temperature, and flexibility compared to traditional serpentine tubes. Pressurization prevents air bubbles from blocking the capillary tubes and assures a constant, uniform flow. 

The “canopy-to-canopy” flow channel was introduced in the research of Mosa, et al. [43] reduces the flow length for better heat transfer and more uniform temperature in comparison with the traditional “serpentine” design, and is recommended for radiant conditioning systems. Hence, the “canopy-to-canopy” flow channel can be applied to the piping layout (Tichelmann system) of a radiant conditioning system, as shown in Figure 8 [14]. This configuration produces the least temperature reduction from entry to exit point, which is very important when operating in the narrow temperature bands of ceiling cooling. Note that cooling is the most challenging to achieve in radiant conditioning, requiring the least temperature difference between supply and ceiling surface temperature.

Later, the “canopy-to-canopy” flow channel was examined in the study of Amanowicz and Wojtkowiak [44] under the name “Z-type structure” and has been proven to be inferior to the “U-type structure”. Here, the study of Amanowicz and Wojtkowiak [44] is conducted with air flow (35,000 L/min) rather than water, and the diameters of the air ducts are from 186 mm to 372 mm, multiple times larger than those used in hydronic radiant systems. Comparatively, the study of Mosa, et al. [43] was specifically conducted for hydronic radiant panels. Also, the “U-type structure” of Amanowicz and Wojtkowiak [44] is unique and has not been seen in any piping design for radiant panels. This “U-type structure” merits further investigation.

### 6.3. Application of These Mechanisms in Hydronic Conditioning Systems

A test chamber (so-called TESTCELL) is used to study and examine the responses of hydronic systems in a near-real-world application (see Figure 9). In this TESTCELL, which was specifically designed with a 75% window-to-wall ratio, this study investigates the effect of radiant ceiling systems within perimeter-zoned office spaces. The experiment setup, along with the measurement and equipment, is described in detail in our previous paper [45]. The facade comprises a commercial-grade clear double-glazed window with a 3-m-high ceiling. Several test runs were conducted to demonstrate the effectiveness of the proposed bioinspired hydronic radiant conditioning system. The test runs are designed to test the thermal performance and capability to provide thermal comfort of the system. The equipment and measurements are shown in Table 3 below:

### 6.4. Performance Evaluation Comparison

The hydronic panel system has several similarities to the human body in terms of energy transfer with its environment. Radiative exchange from the human body, as well as a radiative (hydronic conditioned panel) surface, is between 40–60%. The hydronic panel emits generally about 75–90 W/m^2^ in a heating mode through radiation. The human body accounts for about 60–90 W of heat loss through radiation. Convection over a panel surface can increase energy transfer by 50–80% depending on air velocity. The human body likewise can lose 20–30% more energy due to natural convection and substantially more (50%) when air velocity is significant. Most interesting, however, is the human skin during evaporation processes when a liquid (sweat) changes to a gas (water vapour). Natural human perspiration (a latent energy transfer) at rest in a comfortable environment is between 20–25%. When a conditioned panel operates in cooling mode below a dew point, forming water vapor on its surface, it has been empirically shown to increase cooling capacity between 5–15%, identified as a latent energy transfer. In conclusion, the heat transfer processes of the conditioning of a ‘radiant’ hydronic panel can be considered more efficient when compared to other man-made HVAC systems, mimicking to some degree what occurs in our human bodies.

In a previous study [4], six radiant cooling panels with a capillary tube mat were experimented on. The results show uniform temperatures on the radiant surface, proving the effective heat transfer of the capillary tube mat design. The subject panels are highly responsive with a commencing time constant of less than 10 min. Also, in an experiment, condensation took place on a radiant panel, with a staggering heat flux of 280 W/m^2^, about 45% higher than other tested panels, along with an exceptionally high 13.3 W/m^2^K heat transfer coefficient. The higher coefficient is due to the additional latent heat transfer. This is similar to the effects of sweating and evaporation on the skin to the human body temperature. Studies have been conducted to utilize the benefit of latent heat transfer and evaporation to enhance radiant ceiling thermal performance while preventing condensation. High-tech Nano super hydrophobic surfaces are used as coating for radiant ceiling surfaces to achieve such goals in the studies of Zhong, et al. [10], Liu, et al. [11], and Jiang, et al. [12]. We have introduced a granular-aggregate finishing surface that performs similarly to these materials in preventing droplets from forming. Likewise, our human skin utilizes hair to break up sweat and moisture over its surface.

Detailed experiments and instrumentation applied in the proposed radiant conditioning system are reported in the paper of Do, et al. [45]. Testing the performance of the proposed system in both heating and cooling modes, under conditions of significant solar heat gain, revealed average capacities of 95 W/m^2^ for cooling and 85 W/m^2^ for heating. In cooling mode, the system effectively maintains optimal thermal comfort, as indicated by the Predicted Mean Vote (PMV), with solar heat gains of less than 1.5 kW (or 83 W/m^2^) for an 18 m^2^ area. However, the findings highlight challenges in heating mode, particularly with the radiant ceiling system, leading to significant vertical air-temperature stratification.

To tackle air temperature stratification created by the radiant-heated ceiling, mechanical convective aid is introduced. A ceiling fan, in upward blowing mode and on the lowest setting, was installed in the TESTCELL to provide convective aid to the radiant ceiling. Here, average air velocities of less than 0.3 m/s are measured. Yet, the additional conditioning capacitance is phenomenal, increasing to over 50% from no convective aid. Over 180 W/m^2^ with combined radiation and convection is experienced in cooling mode. In the study of Bauman, et al. [39], a similar experiment was conducted, resulting in about a 22% increase in cooling capacity for the radiant cooling ceiling. However, in this study, the radiant ceiling is not fully exposed to the ceiling fan. Further experiments are required; nevertheless, this indicates the tremendous influence of integrated heat transfer mechanisms, like those occurring in the human body.

## 7. Results and Discussion

### 7.1. Results

#### 7.1.1. Theoretical Framework

The integration of a hydronic radiant ceiling system that mirrors the human body’s circulatory system presents an innovative approach to enhancing indoor comfort. This design emphasizes the importance of both functionality and physiological parallels, aiming to create an environment that is not only efficient but also attuned to human needs. The details can be seen in Table 4 below:

In the human body, the circulatory system uses the heart, arteries, veins, and capillaries for temperature and pressure regulation, expansion, and contraction. The hydronic system utilizes manifolds, main piping, and capillary tubing similarly, and employs a pressure pump to regulate temperature, with a pressure vessel maintaining a pressure of 1.0–2.0 bar.

Both systems regulate the skin’s surface temperature through the circulatory system. The skin, through the blood, and the hydronic system, through the changing (mixed) water supply temperature. Heat transfer regulation in both systems is through self-regulation in tune with the environmental temperature. The body releases, or gains heat from surfaces and air temperatures. In the hydronic system, the energy transfer depends on temperature differences between the radiant surfaces and the room’s operative temperature.

Thermal comfort in the natural system maintains a body temperature of 36 degrees C. The Hydronic system uses an Adaptive Model of Comfort [46], which implies that humans dress and acclimate according to the external environment. A control system based on the Adaptive Model of Comfort using room operative temperature and external mean outdoor air temperature. Perspiration and condensation in the natural system use evaporation as a cooling mechanism. A similar process in the Hydronic system is challenging to achieve; condensation concerns have led to the development of a ‘dewpoint extension’ substrate material. A granular rough-edged surface that prevents water droplets from forming, creating a vapor film to avoid mold growth. Finally, the skin’s cooling in response to air movement at the skin addresses the evaporation of sweat in the natural system. Hence, the hydronic system includes integration with a ceiling fan at low velocity, which increases cooling capacity by 50–100%.

#### 7.1.2. Validation

The hydronic panel system has several similarities to the human body in terms of energy transfer with its environment. Radiative exchange from the human body, as well as a radiative (hydronic conditioned panel) surface, is between 40–60%. The hydronic panel emits generally about 75–90 W/m^2^ in a heating mode through radiation. The human body accounts for about 75–90 W of heat loss through radiation. Convection over a panel surface can increase energy transfer by 50–80% depending on air velocity. The human body likewise can lose 20–30% more energy due to natural convection and substantially more (50%) when air velocity is significant. Most interestingly, however, is the human skin during evaporation processes when a liquid (sweat) changes to a gas (water vapour). Natural human perspiration (a latent energy transfer) at rest in a comfortable environment is between 20–25%.

Figure 10 and Table 5 describe the radiant system and experiment, along with the equipment involved. When a conditioned panel operates in cooling mode below a dew point, forming water vapour on its surface, it has been empirically shown to increase cooling capacity between 5–15%, and also to provide a latent energy transfer. In conclusion, the heat transfer processes are more biomimetic for the radiant hydronic systems when compared to other man-made HVAC systems.

### 7.2. Discussion

In summary, the proposed hydronic radiant ceiling system not only aligns with physiological principles but also enriches our understanding of thermal comfort experiences. By bridging technology with natural human sensations, this system holds the promise of creating more comfortable and energy-efficient spaces for both living and working. The interconnection of radiant and convective methods further supports the notion that a multifaceted conditioning approach to thermal comfort is crucial for optimizing indoor environments.

Further research is warranted to investigate the integration of hydronic cooling systems in buildings. The concept of the “Third Skin” offers an intriguing framework for examining hydronic systems from a biomimetic perspective, facilitating their integration. By drawing parallels between human biology and architectural systems, we can explore innovative methodologies for enhancing existing hydronic designs.

Historically, the concept of skin encompasses layered meanings, extending from our biological protective barrier to the clothing and shelters that humanity has created. In ancient civilizations, both clothing and architecture served as markers of identity, reflecting societal structures, power dynamics, and personal expression. For instance, the Romans utilized clothing to signify social status, while ancient Egyptians adorned themselves with elaborate attire to convey divine authority. By framing buildings as extensions of human experience—a “third skin”—architects like Vitruvius articulated the need for structures that provide not only shelter but also personal and cultural significance. In contemporary architecture, the “Built Environment” has evolved into a dynamic realm where buildings function as an additional layer of protection. The works of modernist architects, such as Le Corbusier and Walter Gropius, highlight the evolution of architecture towards a more expressive and functional role, emphasizing the interplay between inhabitants and their environments. This evolution leads to a refined understanding of how architectural forms can meet the diverse needs of individuals within those spaces. Recent research indicates that early modernists, such as Frank Lloyd Wright, observed connections in their design processes to biomimetic, using environmental analogies to inspire their designs [47].

Expanding on the concept of three layers, the theory introduces the idea of a “fourth skin,” emphasizing the interwoven nature of urban environments where architecture, human activity, and the natural environment coexist harmoniously. The potential for a biomimetic approach in this context is substantial, as nature has perfected numerous processes that could inform and inspire sustainable solutions in architectural and urban design. At the heart of this study is the interaction between human biology and the hydronic systems integrated into our built environments. By analyzing the operational principles of the first skin, we can enhance and develop an active hydronic system that emulates these natural functions. This analogy provides four key concepts:Filtering: Just as human skin regulates the variations between the internal and external environment, hydronic systems filter and adapt to heating and cooling demands effectively.Layered Functions: The multiple layers of human skin performing diverse functions suggest that architectural skins should similarly possess the capability to multitask, thereby enhancing environmental performance.Permeability: The properties of human skin, which facilitate the exchange of nutrients and gases, can inform the design of hydronic systems. Ensuring sufficient surface area for thermal exchange will contribute to more efficient and responsive systems.Connectivity: The interconnectedness observed in skin underscores the importance of integrating with the surrounding environment. While hydronic systems aim for efficient energy transfer, fostering meaningful connections with their context is also essential [48].

Additional research is needed to compare radiant and convective systems. HVAC systems are fundamentally convective systems because they use air movement to transfer heat throughout a building. While HVAC systems also use other heat transfer methods, such as conduction and radiation, convection is the primary mechanism for distributing heated or cooled air via fans and ductwork. A review by Fu et al. suggests that as the demand for improved indoor environmental quality (IEQ) and energy efficiency increases, various nature-inspired cooling technologies aim to enhance the comfort and productivity of building occupants. Maintaining good IEQ is crucial for health and productivity; however, achieving it often requires continuous air conditioning, which can lead to high energy consumption, particularly for cooling.

Hence, the authors propose extending the study to compare radiant and convective systems by adapting the Test cell to run in air-conditioning mode, which will serve as a control. The comparative study will also include economic analysis. Lessons from bio-inspired cooling technologies highlight how animals and plants regulate heat, helping balance building designs with the natural environment, enhancing indoor environmental quality (IEQ), and promoting energy efficiency. The discussion covers various building elements, including HVAC systems and different types of building envelopes, as well as the mechanisms of heat transfer—conduction, convection, evaporation, and radiation [49].

## 8. Conclusions

In conclusion, ongoing research into hydronic conditioning systems utilizing biomimetic methodologies represents a significant advancement in integrating technological innovation with sustainable design principles. By leveraging insights from natural systems, this work not only enhances the performance of heat transfer mechanisms but also fosters a deeper connection between architectural systems and their environments. Specifically, it highlights the importance of understanding infrastructure, particularly the Hydronic Operation and Distribution Module (HODM), for advancing biomimetic artificial hydronic systems. The HODM, along with the instrumentation and continuous development of central algorithms—such as adaptive comfort control—contributes to further advancements in systems that mimic nature more effectively.

The proposed system was tested in both heating and cooling modes under high solar heat gain conditions, demonstrating average capacities of 95 W/m^2^ for cooling and 85 W/m^2^ for heating. In cooling mode, the system maintained optimal thermal comfort with solar heat gains below 1.5 kW for an 18 m^2^ area. In contrast, challenges arose in heating mode, mainly due to significant air-temperature stratification. Previous research on radiant cooling panels with capillary tube mats has demonstrated effective heat transfer and improved performance enabled by latent heat transfer. Studies have focused on enhancing radiant ceiling systems by applying nano superhydrophobic coatings to prevent condensation, achieving moisture management performance comparable to that of human skin.

Future investigations will explore the concept of the building as a “third skin” and the application of eco-mimicry, highlighting the potential of these systems to adapt and thrive within their ecological contexts. Ultimately, this approach advances the capabilities of hydronic conditioning systems and contributes to the broader goal of sustainable development, paving the way for innovative solutions that harmonize functionality with environmental stewardship.

Further experimentation and exploration of biomimetic examples may enhance our understanding of how nature manages temperature differences and extremes to maintain a comfortable equilibrium.

Numerous recent papers emphasize innovative approaches to thermal comfort that prioritize both energy efficiency and user satisfaction. Additionally, an underrepresented yet innovative technology in the literature is wall heating panels with heat pipes. This radiant heating system combines various heat transfer mechanisms. The principle involves extracting heat from water by evaporating the working fluid in heat pipes, with the panel’s surface heated by the vapor’s condensation. This mechanism has parallels in nature. Identifying these natural mechanisms can enhance the discussion and highlight intriguing connections between engineered systems and nature. This approach emphasizes the potential of bio-inspired designs to create innovative solutions for sustainable heating technologies.

## Figures and Tables

**Figure 1 biomimetics-10-00843-f001:**
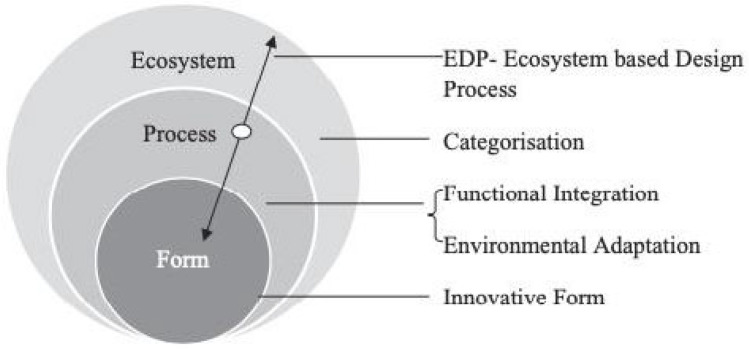
The Model of Biomimicry [22].

**Figure 2 biomimetics-10-00843-f002:**
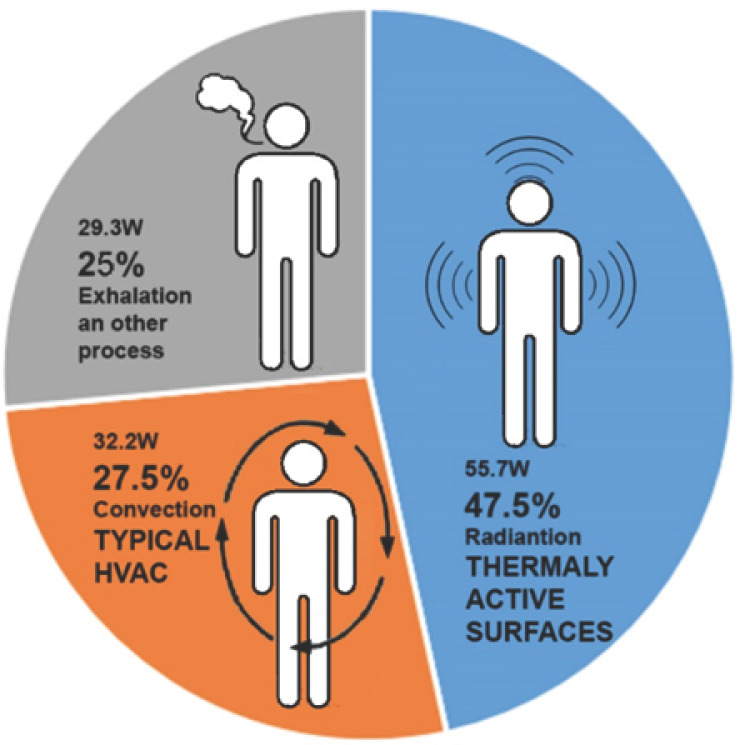
Mechanisms of Heat Transfer on the Human Body (Revised from Moe [23]).

**Figure 3 biomimetics-10-00843-f003:**
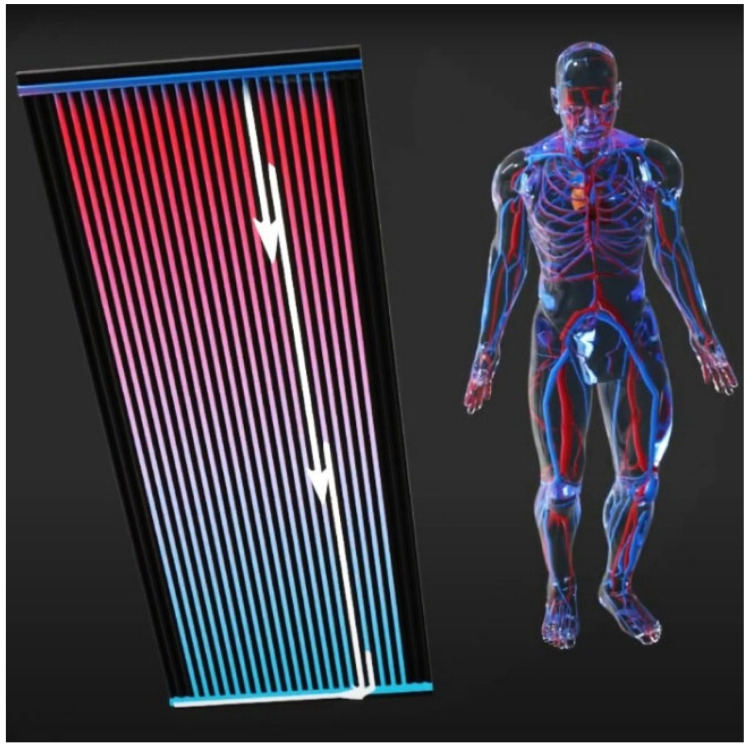
Analogies of Circulation with the Human Vascular System.

**Figure 4 biomimetics-10-00843-f004:**
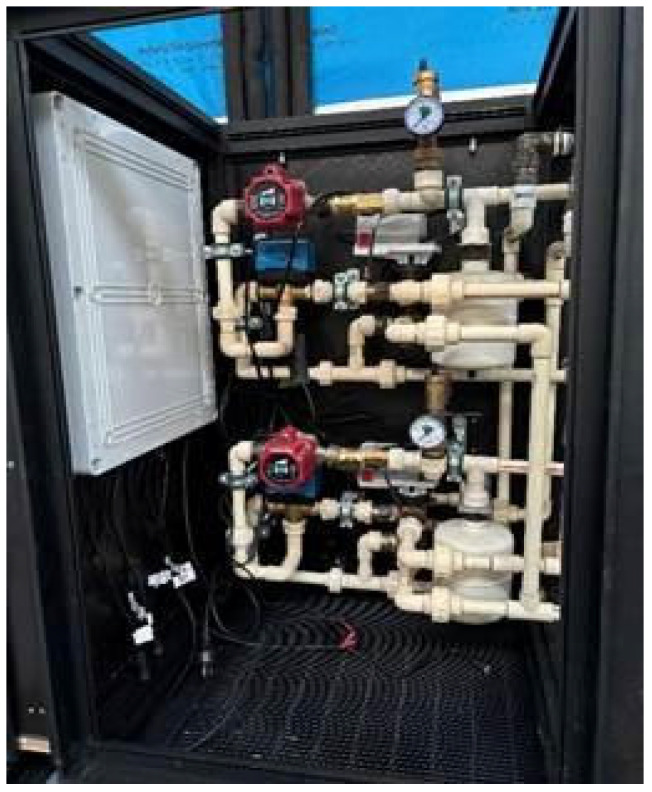
The Hydronic Operations Distribution Module within a Cabinet.

**Figure 5 biomimetics-10-00843-f005:**
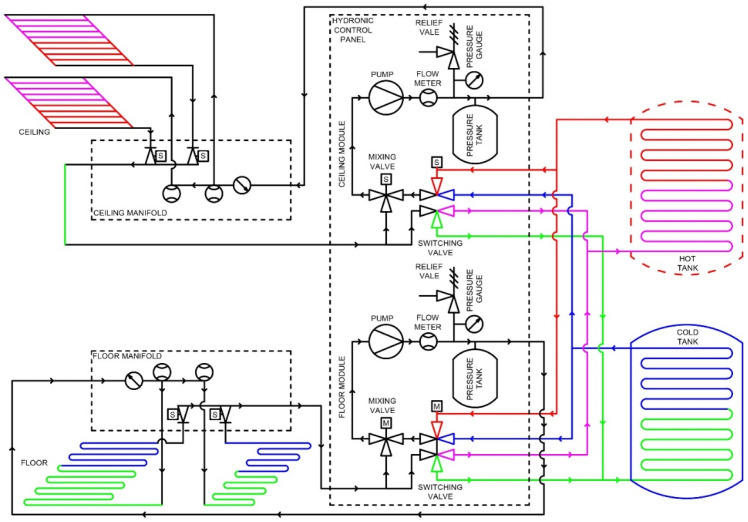
Test-cell radiant system.

**Figure 6 biomimetics-10-00843-f006:**
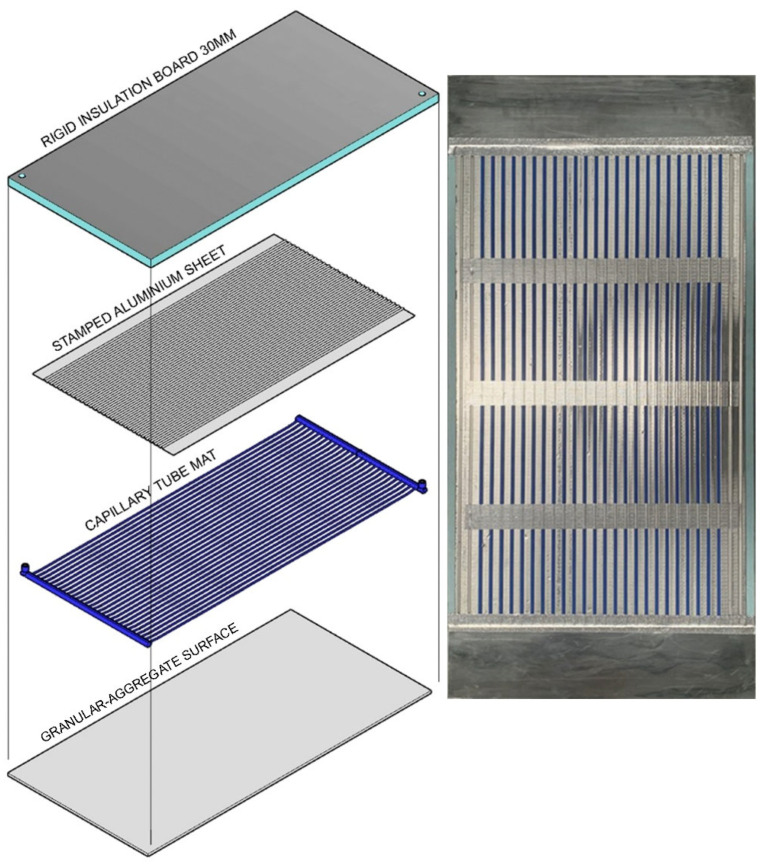
The Capillary Hydronic Panel (minus the finishing substrate).

**Figure 7 biomimetics-10-00843-f007:**
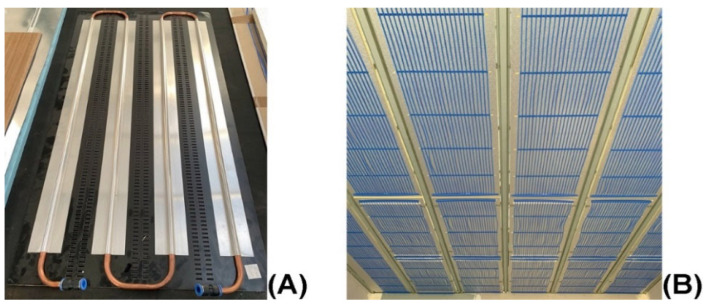
Serpentine (**A**) vs. Capillary Tube Mat (**B**) Design.

**Figure 8 biomimetics-10-00843-f008:**
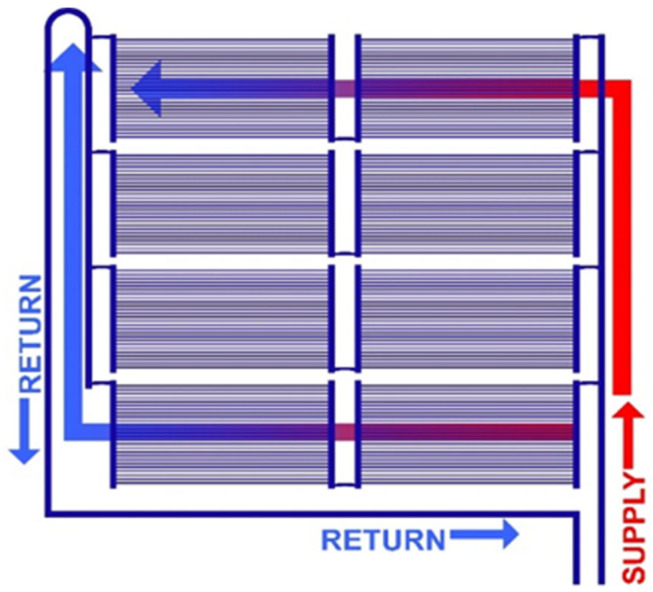
A ‘Canopy-to-canopy’ flow configuration for capillary panels.

**Figure 9 biomimetics-10-00843-f009:**
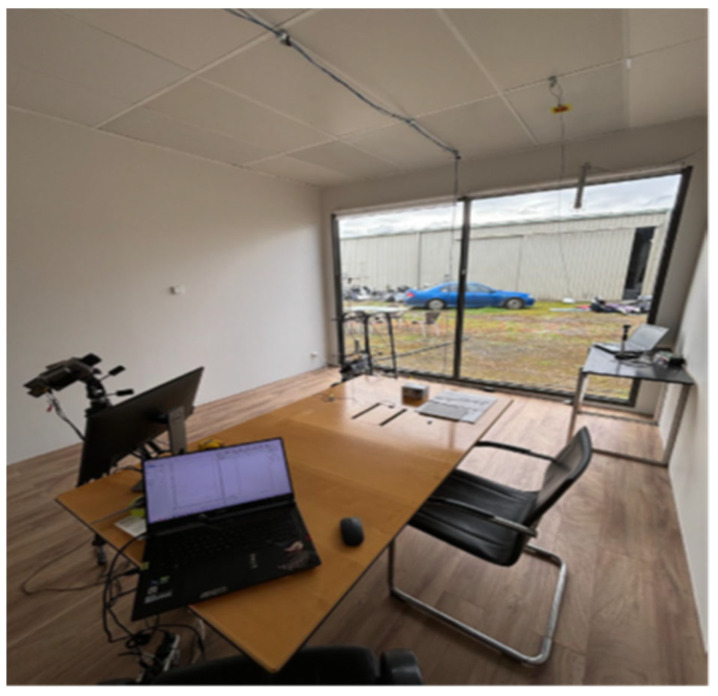
The TESTCELL research and demonstration Room.

**Figure 10 biomimetics-10-00843-f010:**
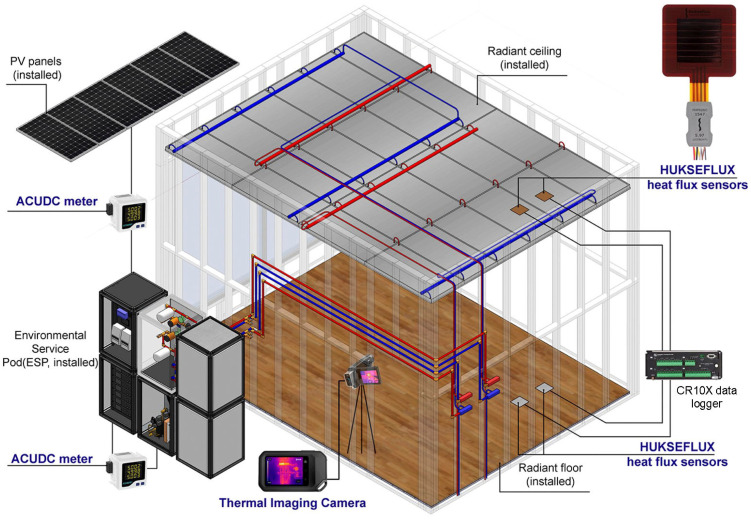
Radiant floor and ceiling monitoring arrangement.

**Table 1 biomimetics-10-00843-t001:** Biomimicry theoretical framework [22].

Scale of Application	Design Process	Direct Approach: Specific Mimicking	Indirect Approach: General Mimicking
Ecosystem: *How does it fit with the whole?* →	Categorization: What is the type of classification? ↓	Type of species, physical characteristics, climatic zones, ↔ Relationship between species, size, and form variations	Identification of building type, types of users, size variations, form variations, relationship with users and organisms, and climatic zones
↓↑ Process: → *How does it perform, and how is it made?* ↓↑ →	Functional integration –What are the innovative strategies? ↓	Hierarchy of functions: primary, secondary, techniques, physical characteristics, ↔ Mechanisms, Patterns, behavioural patterns, needs, communication, organization	Users and user needs, hierarchy of functions: primary, secondary functions, techniques, physical characteristics, mechanisms, user behaviour, patterns, needs, occupancy, communication
	Environmental Adaptation –what are the innovative strategies? ↓	Macro and micro ↔ environment, physical characteristics, habitat, topography, macro and micro climate: wind, sun path, temperature, humidity, rainfall	Macro and micro-environment, physical characteristics, habitat topography, macro and microclimate: wind, sun path, temperature, humidity, rainfall
Form: → *What is the shape?*	Innovation of form—what is the expression?	Design fundamentals: lines, shape, texture, colour, patterns, geometric progression: module, unit to whole, scale, and proportions	Design fundamentals: lines, shape, texture, colour, patterns, geometric progression: module, unit to whole, scale, and proportions

**Table 2 biomimetics-10-00843-t002:** Comparative analysis of analogical translation systems [22].

Analogical Translations	Nature Studies Analysis	Typological Analysis	Design Spiral	Bio-TRIZ
Biomimicry scales of application,	Form, process, ecosystem,	Organism behaviour, ecosystem,	Form, process, ecosystem System, sub-system,	System, sub-system,
Ecosystem: *How does it fit with the whole?* Process: *How does it perform, and how is it made?*	Categorization—scientific reasoning, functional adaptation, contextual adaptation, aesthetic reasoning	Form, process, material, construction, function	Identify, interpret, discover, abstract, emulate, evaluate	Problem, problem understanding, logical solutions
Form: *What is the shape?*	Natural systems to built systems and built systems to Natural systems?	Biology influencing design and design looking to biology	Biology investigating design and design investigating biology	Solution-driven approach and problem-driven approach

**Table 3 biomimetics-10-00843-t003:** Equipment and measurements [45].

Areas	Instrument/Sensor	Measurement/Usage
Weather station	Wind speed anemometer Global solar irradiance sensor Relative Humidity sensor Dry-bulb temperature sensor	External wind speed (m/s) External solar radiation (W/m^2^) External humidity (%) External air temperature (°C)
Window	Solar Irradiance sensors	Solar radiation received at the window (W/m^2^) Solar radiation transmitted through the window into the chamber (W/m^2^)
Thermal comfort cart	CO_2_ sensor Relative humidity sensor Globe thermometers Anemometers Thermistors	CO_2_ level (ppm) Relative Humidity (°C) Mean radiant temperature calculation (°C) Air velocity (m/s) Air temperature (°C)
Chamber, floor, and ceiling	Thermocouple Heat flux sensor Stratification thermocouples Thermal imaging camera	Radiant surface temperature (°C) Energy emission/energy transfer (W/m^2^) Air temperature stratification (°C) Chamber surface temperature (°C)

**Table 4 biomimetics-10-00843-t004:** Human Body vs. Capillary Hydronic System.

	Human Body—Natural System	Capillary Hydronic System
Circulatory system	Heart, Arteries, Veins, and Capillaries, Pressure regulation, expansion, and contraction.	Pump with regulated mixing temperature, Pressure vessel maintaining 1.0–2.0 bar, Manifolds, Main Piping, and Capillary tubing.
Surface Temperature	Regulated Skin Temperature via the circulatory system	A very shallow (minimal depth) of less than 5 mm is the “regulated skin” of the insulated panel. It is controlled via the changing (mixed) water supply temperature.
Heat transfer regulation	A self-regulated system in tune with its environmental temperature. The body releases, or gains heat from, surfaces and air temperatures.	Self-regulation of the energy transfer depends on temperature differences between the radiant surfaces and the room’s operative temperature.
Thermal comfort	Adaptive Model of Comfort implies that humans dress and acclimate according to the external environment.	A control system based on the Adaptive Model of Comfort using room operative temperature and external mean outdoor air temperature.
Sweating and condensation,	Skin sweating and evaporation as a cooling mechanism	Condensation concern has led to the development of a ‘dew point extension’ substrate material. A granular rough-edged surface that prevents water droplets from forming, creating a vapor film.
Air movement Dynamic response	Cooling in response to air movement at the skin. Evaporation of sweat. A fast responding system to changing thermal environments.	Air movement via the system integration with a ceiling fan at low velocity increases cooling capacity by 22% (can be higher). A dynamic response to thermal changes is combined through light-weight radiant panels (<10 min time constant), sensors, and an adaptive control system.

**Table 5 biomimetics-10-00843-t005:** Radiant system, equipment, and measurement.

Item of Interest	Equipment Implemented	Description: Instrumentation/Control
Circulatory System	Pump, Mixing Valve, Flow Sensor, 6-way switching valve, Pressure Vessel	Variable Speed Pump: manual control 0–10 V control of valve opening/closing Mixing Return/Supply water Flow Rate: L/minute Temperature Sensor Pressure Guage
Panel ‘Skin’ Radiative Temperature	Temperature Sensor	Thermocouple/thermistor
Panel ‘Skin’ Energy Transfer	HFM Sensor	Heat Flux Meter (W/m^2^)
System Heat Transfer Regulation	Sensors with Control Program	System Supplied Water (thermistor) DB Room Air Temperature (thermistor) Panel Surface Temperature (thermistor) Heat Flux Meter (W/m^2^)
Adaptive Comfort Control	Sensors with Control Program	Average External (Outside) Air (thermistor) Indoor Operative Temperature: calculated (MRT & DB) PLC—Controller Hardware
Room Air Velocity Control	Variable Speed Ceiling Fan & Air Velocity Sensor	Variable Speed Fan—measured speeds with 6 room air velocity meters to obtain average air movement in the room
Thermal Comfort	Thermal Comfort Cart with Sensors and Data Logger	An ISO 7730 2005-11-15 [41] Comfort Index as calculated by the measured parameters of Air DB, Relative Humidity, Globe Temperature (MRT), Air Velocity, Clothing Index, and Metabolic Rate
Panel Response Time	Temperature Sensors and Data Logger	Water Supply Temperature Panel Surface Temperature Heat Flux Meter

## Data Availability

The original contributions presented in this study are included in the article. Further inquiries can be directed to the corresponding author.

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
