# Peer review of "The Third Skin: A Biomimetic Hydronic Conditioning System, a New Direction in Ecologically Sustainable Design"

_biomimetics, 2025, doi:10.3390/biomimetics10120843_

Round 1

Reviewer 1 Report

Comments and Suggestions for Authors

The main goal of this paper is to explore how biomimetic principles inspired by the human circulatory and skin systems can be applied to improve hydronic radiant conditioning systems in buildings. By integrating biological analogies into engineering design, the research aims to create more sustainable, energy-efficient, and responsive systems that can maintain indoor comfort while minimizing environmental impact. The following revisions are recommended to improve the overall quality of the paper:

1- The abstract provides a general overview but lacks quantitative results. It should include key numerical findings to strengthen the scientific depth.

2- The conclusion is coherent but rather general. It should summarize key quantified outcomes and specific implications for practical application or policy development.

3-  It is recommended that the Introduction section be presented as a single, continuous narrative rather than being divided into multiple subsections. The existing subheadings should be removed, and the text should appear as an integrated discussion under the main title “1. Introduction.”

4- It is suggested that Section 2 be renamed “Methodology”, and that the explanations of the conceptual framework currently presented elsewhere be included within this section to create a clearer and more coherent structure.

5- With regard to validating the proposed theoretical framework in the context of building conditioning systems, the manuscript would benefit from the inclusion of additional quantitative data and supporting analysis in the Results and Discussion section to substantiate the theoretical claims.

Reviewer 2 Report

Comments and Suggestions for Authors

The paper is about the use of biomimetics in thermal comfort systems (i.e. radiant hydronic heating). The objective of the paper is “to use biometric methods to provide insight into potential improvements of the systems and a testing methodology”. Paper presents an interesting, innovative approach that could introduce a new perspective on thermal comfort and the design, operation, and operation of HVAC (heating, ventilation, and air conditioning) systems. The work is not innovative and does not contribute new knowledge. Rather, it organizes existing knowledge and many years of experience in the use of radiant heating systems. The only interesting aspect is the comparison of heat transfer processes in heating systems to heat transfer processes in the human body. This reinforces and strengthens the foundations of knowledge about these systems, which are already popular and widely used, as evidenced by the large number of studies in this area (thermal comfort and HVAC systems), which the authors unfortunately did not access and do not cite. Hence, the belief that the topic is new, even though it is not. Nevertheless, the topic is described in a different way than previously encountered, which, to a minimal extent, still meets the requirements for publication, in my opinion. The work is interesting/ Nevertheless, below I suggest some changes that will improve the article from the perspective of an HVAC engineer.

1) Please rewrite to abstract in the following manner: (i) Highlight the background and the purpose of the study, (ii) describe briefly the main methods used to achieve the goals, (iii) Summarize main findings and conclusions by highlights the great contribution to the field. Quantitative findings rather than qualitative ones are highly recommended. At his moment the biggest gap can be noticed in the scope of the method, the description of the scope of work (what exactly does it concern?) and the conclusions, observations, even a preview of the most interesting results or a description of the most important achievements of this work.

2) Literature review is quite short. From my point of view it would be better to refer to more actual papers showing actual achievement in energy and comfort systems. Another suggestion – I can find a gap in not popular but present int the literature radiant system: wall heating panel with heat pipes. It's interesting to describe it in the context of the introduction, where you write about various forms of heat transfer. Here, it's interesting that heat is extracted from water by evaporation of the working fluid in the tubes and heating of the surface panel by the tube walls, which are heated by the condensation of the working fluid vapor. Could you try to find such a mechanism in nature? Or perhaps identify similarities and differences? I would find it as very interesting.

3) Please describe the methods in a way that enable repeating of your investigations. Please provide a table with the list of measurement equipment used in the experiment with description and its accuracy. What is more please clearly show on the schema of the experimental set the location of the measuring equipment. Please prove that all the equipment was used properly, i.e. the beginning sector for flow development were kept (if the flow was measured) etc. Every information that is needed to repeat your investigations should be provided.

4) Conclusions should be improved. Please add more quantitative conclusions. Please highlight the novelty and great meaning of the work in the field. Please better describe the next steps. Please refer your results with results of previous papers from other authors – there are a lot of papers about thermal comfort and radiative HVAC systems.

Round 2

Reviewer 2 Report

Comments and Suggestions for Authors

The authors have significantly improved parts of the article.

The issue of experimental research remains – only a photo of the hydronic system is shown, without any information about what was measured, what devices were used, or to what accuracy. The photo of the pipe braid (Fig. 4) does not clarify these issues. An international article should include all the information necessary to fully replicate the research. I strongly recommend correcting this.

Fig. 7 "Canopy-to-canopy" is not a technical term. In engineering, this structure is called a Tichelmann system, ensuring the equal length of each possible flow path. You can find articles that refer to it with a letter – a "Z" structure, a Z-type structure – for example, see the U and Z structures for connecting multi-pipe earth-to-air heat exchangers: https://doi.org/10.1016/j.geothermics.2020.101896.

Generally, the systems you describe are not new and are already widely used in buildings. Only in low-budget investments do other solutions apply, but this is not due to a lack of knowledge, but rather a lack of money. Therefore, I suggest rereading the issue of "novelty" and the "knowledge gap" to avoid artificially labeling systems as innovative when they are not. Instead, focus on the truly interesting aspect of the article, which explains in greater detail and better than before the relationship between HVAC systems, which provide thermal comfort because they operate in a manner analogous to the heat exchange processes between humans and their surroundings. This better explanation of the reasons why these systems result in improved comfort is a strength of this work.
